# Assessing the Bioreceptivity of Biobased Cladding Materials

**Karen Butina Ogorelec** [1,2,*], **Ana Gubenšek** [1,2], **Faksawat Poohphajai** [1,2,3] **and Anna Sandak** [1,2,4,*]

1   InnoRenew CoE, Livade 6a, 6310 Izola, Slovenia; ana.gubensek@innorenew.eu (A.G.); faksawat.poohphajai@innorenew.eu (F.P.)
2   Andrej Marušič Institute, University of Primorska, Muzejski trg 2, 6000 Koper, Slovenia
3   Department of Bioproducts and Biosystems, Aalto University School of Chemical Engineering, 00076 Aalto, Finland
4   Faculty of Mathematics, Natural Sciences and Information Technologies, University of Primorska, Glagoljaška 8, 6000 Koper, Slovenia
*   Correspondence: karen.butina@innorenew.eu (K.B.O.); anna.sandak@innorenew.eu (A.S.)

**Abstract:** Materials exposed to the outdoors are prone to various deterioration processes. Architectural coatings are designed to protect surfaces against environmental and biotic degradation and to provide a decorative layer. The objective of this work was to examine the early colonisers on a diverse set of coated and non-coated biobased façade materials. A set of 33 wood-based cladding materials were exposed to four cardinal directions and monitored in outdoor conditions. The surfaces were sampled using a wet swab and plated on DG-18 agar, which prevents the growth of bacteria and limits the growth of fast-growing fungi. Pure cultures were then isolated and identified through PCR amplification and Sanger sequencing of specific DNA regions/genes. The response of cladding materials to weathering and fungal infestation was assessed. The proposed techniques enabled the identification of features that promote/inhibit fungal colonisation and revealed the preference of certain fungi for specific materials. Both the material type and the climate condition at the exposure site influence fungal colonisation. This study is a starting point for more exhaustive assays that aim to develop a novel coating system based on controlled and optimized fungal biofilm formation, and is proposed as a nature-inspired alternative for the protection of architectonic surfaces.

**Keywords:** coating; materials performance; bioreceptivity; natural weathering; fungal infestation; early fungal colonisers

## 1. Introduction

Wooden cladding materials are susceptible to biotic attack by moulds and decay fungi. The latter causes structural weakening of the wooden material, while the former leads to surface discolouration and affects the aesthetic appeal. The occurrence of either type depends on the environmental conditions, wood moisture content, and durability of the cladding material [1]. Together with other microorganisms and insects, fungi contribute to the biodeterioration of wood that occurs in parallel with abiotic deterioration caused by wind, rain, temperature, and UV radiation.

While their aesthetic appeal, availability, affordability, and environmental friendliness contribute to the popularity of wooden cladding materials, their maintenance remains challenging due to the reasons stated above [2]. Coatings are often applied for wood protection but have historically contained substantial amounts of VOCs. With growing environmental concerns, the use of high-VOC paints has been highly restricted according to European Directive 2004/42/CE. Low-emission coatings are, however, still niche products since 70% of the raw materials are based on petroleum (mineral oil) [3]. Nevertheless, there is a shift from solvent-based to water-based architectural coatings. Water-based coatings unfortunately do not always match the performance of solvent-based ones [4], posing a significant challenge to the coating industry. The water-based formulation leads to quick

evaporation of water resulting in tacky, more viscous paint [5]. Architectural coatings manufacturers are, therefore, focusing on developing formulations that offer longer open times for paint users, defined as the "time after application beyond which further reworking of the paint film results in visible surface defects" [6]. Moreover, aqueous raw materials used in the formulation of paints and coatings create a beneficial environment for the growth of bacteria, fungi, and yeast [7]. To prevent the growth of microorganisms, various biocides can be employed. Copper-rich systems are commonly used; however, there are concerns regarding damage to aquatic systems as well as the disposal of treated wood [8]. Due to the stringent EU legislation as well as awareness of health and environmental issues, there is a need to develop non-toxic biobased wood coatings with satisfactory performance [9].

Bioreceptivity is defined as the ability of a material to be colonised by one or several groups of living organisms without necessarily undergoing any biodeterioration [10]. However, it is often desirable to remove any kind of colonisation from building surfaces for preventive conservation but also to maintain cleanliness and order [11]. Nevertheless, in line with the bioreceptivity concept, an innovative approach to wood protection has been described, where the growth of a yeast-like fungus, *Aureobasidium pullulans* (*A. pullulans*), on wood impregnated with linseed oil has exhibited protection of the wood rather than its degradation [12]. This was made possible by understanding the interaction between the wood and the fungus and creating circumstances where the fungus exhibited a protective effect. We consider this an important milestone in the development of alternative coating systems, demonstrating that fungi are not necessarily only harmful but can be beneficial as well. An understanding of fungal colonisation on façade materials is indispensable to further such efforts and develop novel concepts for material protection. An example of such an alternative coating system based on a controlled and optimised fungal biofilm is currently under development within the frame of the ARCHI-SKIN project [13].

In most studies where fungal growth on wooden materials was investigated, the earliest time point for fungal infestation evaluation is 3 months after sample exposure [14–16] or more [17,18]. We would like to argue that earlier time points deserve more attention as they can provide information about the initial steps of fungal attachment and growth on surfaces [19], which is relevant both for their prevention and promotion.

In the literature, samples are often exposed in a single cardinal direction [15–18], yet Podgorski et al., who exposed samples in all four cardinal directions, found they did differ [14]. In real life, cladding materials are exposed in all cardinal directions (on buildings) and the information about their respective performance is of relevance. Moreover, the architectonic design and specific 3D geometry of buildings create specific zones that might be more or less exposed to the weathering process. Consequently, environmental exposure (dose) is closely related to material response [20].

The objective of this work was to examine the early colonisers on a diverse set of façade materials including commercial materials that are coated and non-coated. The effect of the exposure site on the microbial burden and the presence of dominant species on various materials were investigated. Simultaneously, the response of cladding materials to weathering and fungal infestation was assessed to determine whether fungal presence is necessarily connected with material deterioration. The experiment is ongoing with the aim of identifying fungal species that simultaneously possess the potential to uniformly cover the surface and not cause material deterioration.

## 2. Materials and Methods

### 2.1. Experimental Materials and Natural Weathering

Experimental samples were obtained from 30 industrial partners from 17 countries that provided experimental material for the BIO4ever project [21] and included specimens representing natural wood, thermally, chemically, and surface-treated wood, impregnated wood, biobased composites, and hybrid modification merging more than one of the above categories. The size of each individual sample was ~150 L × ~75 W × ~20 T mm$^3$. Before

weathering, samples were scanned with an office scanner HP Scanjet 2710 (300 dpi, 24 bit) and saved as TIF files (Figure 1a). All information available about the samples can be found in Table 1. The 33 different cladding materials analysed in this work were a subset of 93 different cladding materials that were placed on the roof of the InnoRenew CoE building, situated in the coastal town of Izola, Slovenia (45.5350, 13.6577). For each cladding material, 2 duplicate samples (left, right) were exposed in each cardinal direction. Samples were mounted on a vertical stand on 13–14 October 2022 and exposed to natural weathering. The results related to microbial growth reported in this work are from observations conducted after the first month of exposure, unless when explicitly stated otherwise. The meteorological data were obtained from the Historical Weather API (https://open-meteo.com/, accessed on 14 June 2023) (Figure 1b). The data are based on reanalyses of datasets and employ a combination of radar, satellite, aircraft, buoy, and weather station observations. Mathematical models are used to fill in gaps in the data. Reference (=unweathered) samples were stored indoors at ambient conditions (~50% RH, ~21 °C).

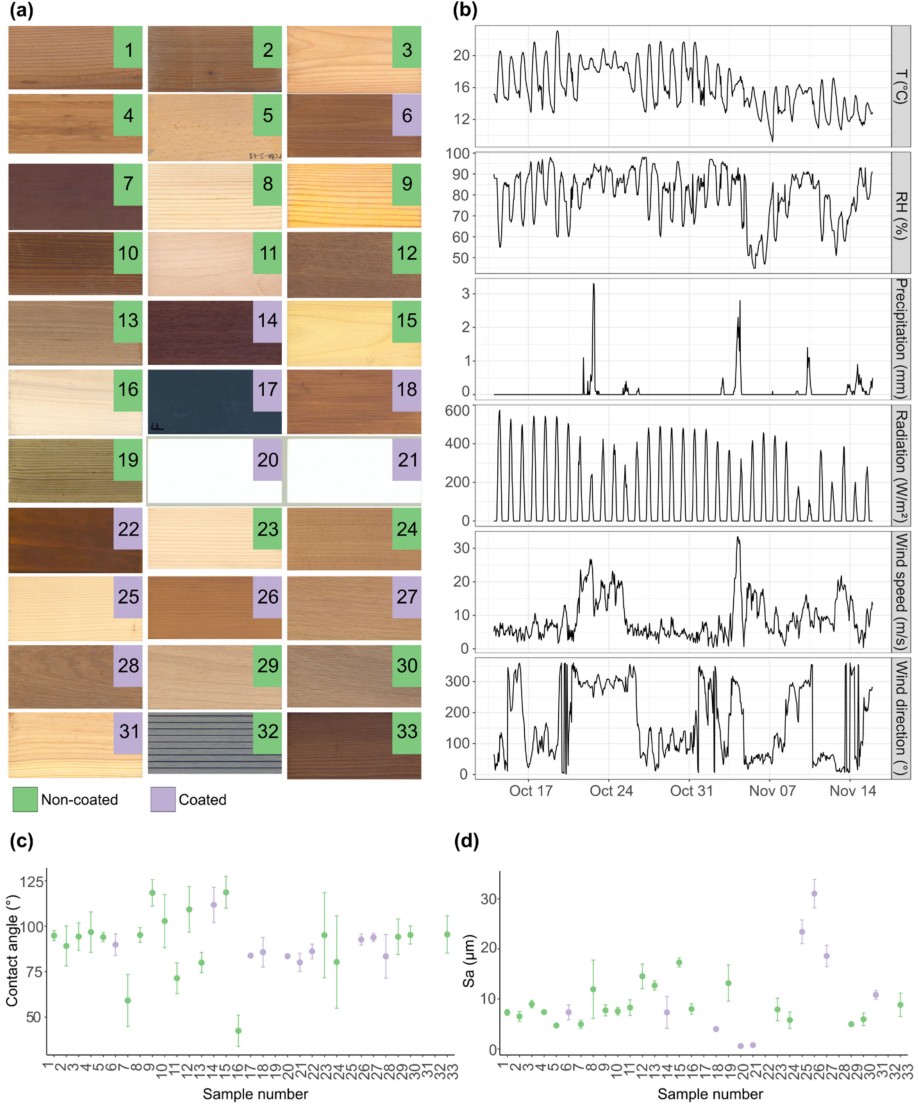

**Figure 1.** (**a**) Scans of samples included in the study. The sample number (see Table 1) in the purple square represents coated samples while those in the green square are not coated. (**b**) Weather data for one month of exposure (14 October 2022–15 November 2022). Hourly data for temperature (T), relative humidity (RH), precipitation, shortwave radiation (radiation), wind speed, and wind direction are displayed. (**c,d**) Contact angle (**c**) and Sa (**d**) of reference samples. Mean ± SD of technical replicates is shown. Colours are the same as in (**a**).

**Table 1.** List of cladding materials.

| Sample nr. | Group [1] | Sample Information | Coating | Species |
|---|---|---|---|---|
| 1 | therm | Treated in 212 °C | no | spruce |
| 2 | hybr | Treated in 212 °C with addition of $FeSO_4$ | no | spruce |
| 3 | natr | Heartwood kiln dried 50–70 °C | no | pine |
| 4 | surf | Oiled stick-glued bamboo façade profiles | no | bamboo |
| 5 | impr | Wood impregnated with silicone and silicate | no | beech |
| 6 | hybr | Thermally treated and coated (industrially) Semi-transparent coating | yes | spruce |
| 7 | chem | Furfurylated (industrially) | no | pine |
| 8 | impr | Impregnated with $TO_2$ nanoparticles | no | pine |
| 9 | hybr | Impregnated with $TO_2$ nanoparticles and linseed oil | no | pine |
| 10 | hybr | Thermally treated and impregnated with copper oxide, boric acid, and tebuconazole | no | pine |
| 11 | natr | Air dried | no | beech |
| 12 | therm | Hydro-thermally treated (industrially) | no | frake |
| 13 | chem | Acetylated (industrially) | no | beech |
| 14 | hybr | Thermally treated and coated (industrially) Semi-transparent coating | yes | frake |
| 15 | impr | Soaked with concentrated fluorosilane | no | poplar |
| 16 | impr | Melamine treated | no | poplar |
| 17 | comp | Coated acetylated MDF (industrially) Opaque petroleum grey coating | yes | composite |
| 18 | surf | Coated with solvent-based product (industrially) Semi-transparent coating | yes | pine |
| 19 | impr | Treated with copper ethanolamine | no | spruce |
| 20 | comp | Coated acetylated MDF (industrially) Opaque white coating | yes | composite |
| 21 | hybr | Acetylated and coated (industrially) Opaque white coating | yes | pine |
| 22 | hybr | Acetylated and coated (industrially) Semi-transparent coating | yes | pine |
| 23 | natr | Natural fir (kiln dried) | no | fir |
| 24 | therm | Thermally treated fir (vacuum) | no | fir |
| 25 | surf | Coated (industrially)Transparent coating | yes | fir |
| 26 | hybr | Coated thermally treated fir (industrially) Transparent coating | yes | fir |
| 27 | surf | Coated natural oak (industrially) Transparent coating | yes | oak |
| 28 | hybr | Coated thermally treated oak (industrially) Transparent coating | yes | oak |
| 29 | natr | Natural oak (kiln dried) | no | oak |

**Table 1.** *Cont.*

| Sample nr. | Group [1] | Sample Information | Coating | Species |
|---|---|---|---|---|
| 30 | therm | Thermally treated oak (vacuum) | no | oak |
| 31 | surf | Nanoparticle transparent coating, 3 layers (DIY application) | yes | pine |
| 32 | comp | Wood–plastic composite | no | composite |
| 33 | therm | Over-thermally treated spruce | no | spruce |

[1] Group: therm = thermally treated; hybr = hybrid; natr = natural; impr = impregnated; chem = chemically treated; comp = composite; surf = surface treated.

*2.2. Surface Wettability and Topography Measurements*

The contact angle and topography measurements of reference samples were obtained with an Attension Theta Flex Auto 4 optical tensiometer equipped with a 3D topography module (Biolin Scientific, Gothenburg, Sweden). On each sample, five contact angle measurements with distilled water were performed. The drop volume was 4 µL, and the contact angle was evaluated at the 20 s timepoint. Image processing was performed with OneAttension v 4.0.5 software, and the Young–Laplace equation was employed to determine the contact angle. The contact angle could not be measured on samples 19, 25, 31, and 32. The last had a grooved surface, 19 and 25 were too hydrophilic and the contact angle could not be measured for 20 s during all five measurements, while 31 was too hydrophobic and the droplet did not attach to the surface. Three topography measurements, each covering an area of 3.2 mm × 2.8 mm were performed. The Sa parameter was evaluated. Measurements were performed on all reference samples except 17 and 22, which were too reflective, 28, where the signal-to-noise ratio was not acceptable, and 32, due to its particular topography. The contact angle and topography were also measured on a subset of wooden samples dismounted from the roof at the 2.5-month timepoint. The measurements were performed after climatising materials to the equilibrium moisture content (~50% RH, ~21 °C). The procedure was similar to the one described above; however, contact angle measurements were performed in triplicates and not quintuplicates.

*2.3. Evaluation of Microbial Growth*

Dichloran glycerol agar (DG-18, REF: 40587, NutriSelect® Plus, Merck Millipore, Merck KGaA, Darmstadt, Germany), potato dextrose agar (PDA, REF: 4019352, Biolife, Milan, Italy), and malt extract agar (MEA, REF: 4016552, Biolife, Milan, Italy) were prepared according to the manufacturers' instructions and poured into 90 mm diameter plastic Petri dishes. The plates were left to solidify at ambient conditions and stored at 4 °C until use. Nylon swabs (REF: 2123-1003, Citotest, Haimen, Jiangsu, China), wetted in sterilised distilled water, were used for sampling. An area of approx. 2.5 cm × 2.5 cm in the upper left corner of each sample was swabbed and directly spread on DG-18. Plates were incubated under ambient conditions for 1 week before evaluation. Colony forming units (CFU) were counted if possible—100 was defined as the upper limit and all plates containing counts >100 were assigned a CFU count of 100. From DG-18 plates with at least 10 CFU, where at least 90% of the colonies appeared to be of the same morphology (by visual evaluation), a representative colony of the prevalent morphology was isolated to a pure culture on PDA plates. Cultures were deposited in the Microbial Culture Collection Ex (Department of Biology, Biotechnical Faculty, University of Ljubljana, Ljubljana, Slovenia).

*2.4. Visualisation and Identification of Fungal Macro- and Micromorphological Features*

Selected cultures were retrieved from the Microbial Culture Collection Ex and grown on MEA at ambient conditions for 7–9 days. The macromorphology of pure cultures was visualised and imaged using the colonyQuant automated colony counter (Schuett, Göttingen, Germany). For *A. pullulans*, *A. melanogenum*, and *Penicillium* sp., a small portion of each respective pure culture was placed directly in a drop of 0.9% *w/v* sodium chloride

solution on a microscope slide, covered with a cover glass and sealed with nail polish. From pure cultures of *Alternaria alternata* and *Cladosporium* sp., two slide cultures were prepared using MEA agar cubes, as described previously [22]. Briefly, the agar cubes were placed on microscope slides, inoculated with each of the selected pure cultures, and covered with a cover glass. The slide cultures were incubated at 25 °C for 3 days. Subsequently, the cover glass was removed, placed on a drop of 0.9% *w/v* sodium chloride solution on a microscope slide, and sealed with nail polish. Transmitted light microscopy was performed with an EVOS™ M7000 Imaging System (Thermo Fisher Scientific, Waltham, MA, USA) using a 60× objective.

A subset of wooden samples was dismounted from the roof at the 2.5-month timepoint and imaging was performed after climatising materials to the equilibrium moisture content (~50% RH, ~21 °C). The samples were imaged with a Keyence VHX-6000 digital microscope (Keyence, Osaka, Japan). Colour images were acquired at 500× magnification.

### 2.5. Molecular Identification of Selected Isolates

Prior to DNA isolation, selected isolates were grown on PDA plates for one week. Genomic DNA was extracted after mechanical lysis in CTAB buffer as described previously [23]. Identification was based on PCR amplification and Sanger sequencing of certain DNA regions/genes. These included internal transcribed spacers 1 and 2 including the 5.8S rDNA (ITS) with primer set ITS5 [5′-GGA AGT AAA AGT CGT AAC AAG G-3′] and ITS4 [5′-TCC TCC GCT TAT TGA TAT GC-3′] [24] for a majority of isolates, partial sequences of genes encoding for actin (act) for genera *Alternaria* and *Cladosporium* with primers ACT-512F [5′-ATG TGC AAG GCC GGT TTC GC-3′] and ACT-738R [5′-TAC GAG TCC TTC TGG CCC AT-3′] [25], and β-tubulin (benA) for genus *Penicillium* with Ben2f [5′-TCC AGA CTG GTC AGT GTG TAA-3′] [26] and Bt2b [5′-ACC CTC AGT GTA GTG ACC CTT GGC-3′] [27]. After obtaining sequences of isolates from Microsynth (Austria), the most similar sequences of type strains and other important taxonomical reference strains were retrieved from the GenBank nucleotide database with the blast algorithm [28]. All sequences were aligned and phylogenetically analysed using the maximum likelihood method as implemented in program Mega11 [29]. All DNA sequences from the representative isolates from this study were deposited in the GenBank database: OR054020-OR054066.

For two samples where a majority of pink yeast (presumably *Aureobasidium* sp.) was observed, we were not able to obtain the species identity (29 west, left panel and 18 east, left panel).

### 2.6. Data Analysis

The R programming language [30] was used for data analysis. R and Inkscape were used for data visualisation. *p*-values were adjusted using the Bonferroni multiple testing correction method. The size of all tests was 0.05.

## 3. Results and Discussion

### 3.1. Experimental Setup

The investigated sample set consisted of a wide range of coated and non-coated commercial cladding materials (Table 1, Figure 1a). Since most of the samples were obtained from commercial entities, the exact chemical composition of certain coatings and treatments could not be obtained. Nonetheless, we considered these materials relevant for investigation since these are the materials that are used as façade elements in "real life." Just over a third of the investigated materials were coated, half of those were hybrid materials, one-third were surface-treated, and the rest were composites. Among non-coated samples, thermally treated and impregnated were the most represented, followed by natural, hybrid, chemically treated, surface-treated and composites. Eight different species were represented among non-coated samples, five among coated. The materials were exposed in all cardinal directions, with duplicate samples for each material in each direction—all together this amounted to eight samples for each material.

The roughness and wettability of investigated materials were evaluated (Figure 1c,d). We considered these parameters important as it has previously been shown that they influence fungal spore adhesion to surfaces [31,32]. There was no overall difference in the wettability and roughness between the coated and non-coated materials, both categories containing samples that spanned a wide range. Interestingly, certain non-coated samples (7, 10, 12, 23, 24) exhibited a rather wide scatter of the measured contact angles but not of Sa. This might be due to biological nature and intrinsic heterogeneity of investigated materials (e.g., differences between early and late wood, presence of wood defects).

The samples were exposed to natural weathering between mid-October and mid-November (Figure 1b). This was early autumn in a coastal region, with consistently high relative humidity and temperatures between 10 °C and 25 °C. These conditions were favourable for microbial growth [33] and the results presented in this work would most likely be different if another time period would be chosen for weathering and sampling.

### 3.2. Microbial Burden across Cardinal Directions

It has previously been reported that the cardinal direction of exposure affects the colour and chemical changes in the wood due to weathering [19,34–36]. Few reports are available on the effects of cardinal direction on fungal growth and/or associated surface discolouration [14,36]. To investigate this, samples exposed in all directions were swabbed and colony forming units (CFUs) were counted (Figure 2). The placement of sample materials on the stand (Figure 2a) and a visual representation of averaged counts of duplicate samples exposed in all cardinal directions are represented (Figure 2b–e).

It is important to mention that despite the physical contact between the samples, the fungal counts of neighbouring samples can differ profoundly. This means that the fungi did not spread uncontrollably between materials, indicating that their growth was governed by specific material properties.

Differences between the cardinal directions are summarised in a boxplot shown in Figure 2f. For all directions, we have materials that range between 0 and 100 colonies (upper limit); however, the central tendencies (represented by median and mean) between the cardinal directions differ. The distributions of the CFU counts on samples exposed in the four directions were found to differ significantly. The least growth was observed in the south, which might be due to the longer exposure to UV radiation leading to faster changes in surface moisture [19]. The highest fungal burden was observed in the east and west, which can partly be explained by the predominant wind direction in autumn in Izola (Figure 1b). The wind-driven rain is the largest moisture source that influences the hygrothermal performance of building envelopes [37], causing more pronounced mould growth.

Wet swabbing and direct re-plating on agar is a relatively fast method that made it possible to swab the 264 samples exposed on the roof in a single day, but there are shortcomings that need to be taken into account. Since only a limited area of the sample was swabbed, the results might not always be representative of the whole surface. Moreover, the swabbing technique in itself is problematic for the enumeration of CFUs since spores can produce more colonies on agar compared to the same biomass consisting of hyphae, and the choice of culturing media can affect the growth of fungi [38]. We did not perform dilution plating, which resulted in some plates being overgrown and the CFUs could not be enumerated but were assigned the arbitrary number of 100 CFU. Nonetheless, the approach taken was robust and enabled comparisons between samples, since the same errors were introduced for all samples. However, the interpretation of any absolute values should be performed with caution.

These results demonstrate that the cardinal direction of exposure influences fungal colonisation of materials and is in line with previously published reports [39]. The doses of UV, wind, and wind-driven rain all strongly depend on the exposure direction and influence material weathering, which might directly and indirectly influence fungal growth.

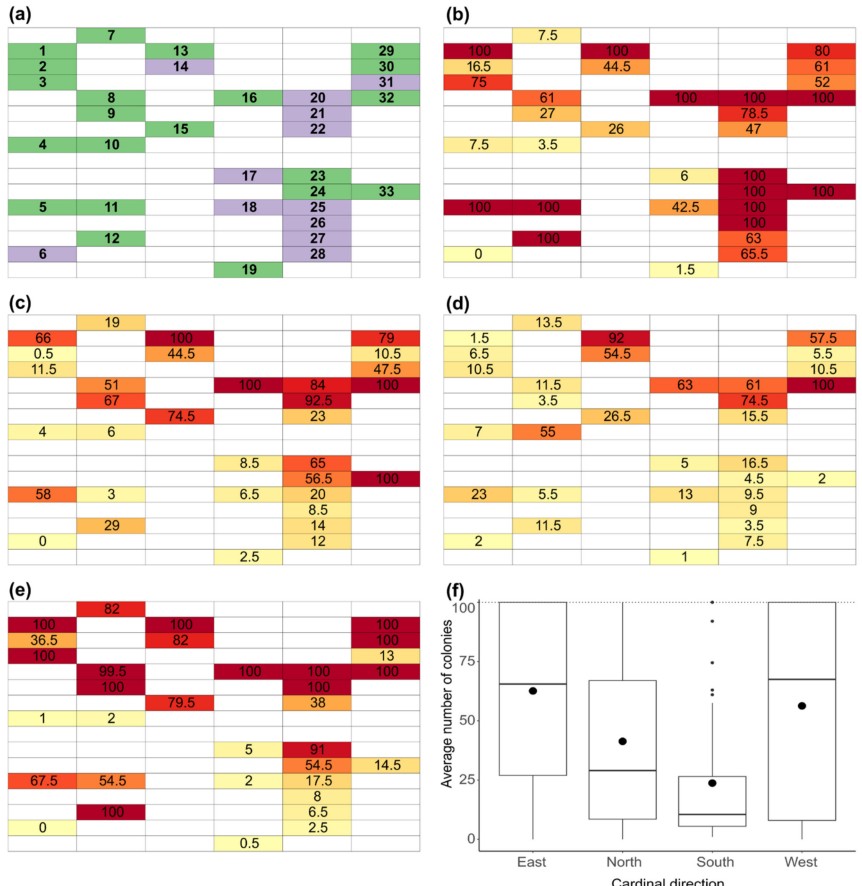

**Figure 2.** (**a**) Placement of material samples on the stands. The sample number (see Table 1) in the purple square represents coated samples while those in the green square are not coated. (**b**–**e**) Average number of colonies from duplicate samples exposed to the (**b**) east, (**c**) north, (**d**) south, and (**e**) west. (**f**) Average number of colonies at different cardinal directions. Black points represent mean values. The four groups do not have equal distributions (Friedman test, $p < 0.05$). There are significant differences between east and north, east and south, north and south, west and south (Wilcoxon signed-rank test, $p < 0.05$). Note: The colour gradient (bright yellow to dark red) corresponds to the average number of colonies (0 to 100, respectively).

### 3.3. Materials Resistant to Fungal Growth

Due to the biodegradation and discolouration associated with fungal colonisation and growth on wooden materials, great efforts are being invested in prevention. Most of the materials used in this study are commercial and several were treated to provide antifungal properties (Table 2).

To determine which samples were not susceptible to fungal colonisation and growth in our test, we identified those that had less than 10 CFUs from the swabbed regions of at least seven out of eight exposed samples (Figure 3a). Two of these samples were coated (samples nr. 6, 17) and two were not (samples nr. 4, 19). Two and a half months after the samples were exposed (that is 6 weeks after the first sampling), several samples (all N-L) were dismounted, conditioned, and examined under the microscope for the visual presence of fungi. Interestingly, samples 4 (Figure 3b) and 6 (Figure 3c) were still devoid of any visible fungal growth. This was in stark contrast with several other samples, where abundant growth was observed. As an example, sample 13 is shown in Figure 3d.

**Table 2.** Cladding materials with putative antifungal properties.

| Sample | Antifungal Treatment/ Property | Mode of Action |
|---|---|---|
| 2, thermally treated and impregnated | Iron(II) sulphate | Excess iron leads to disturbed iron homeostasis [40,41] |
| 5, impregnated | Silicone and silicate | Acts as a water repellent and reduces moisture uptake of wood [42,43]. |
| 8, impregnated 9, impregnated | Titanium dioxide nanoparticles | Generation of free radicals (hydroxyl and superoxide anion) and hydrogen peroxide causing microbial growth reductions [44] |
| 10, thermally treated and impregnated | Copper oxide, boric acid, and tebuconazole [45] | Copper can cause membrane damage [46] and protein denaturation [47]. Tebuconazole leads to ergosterol degradation [48,49]. |
| 15, impregnated | Fluorosilane | Acts as a water repellent and reduces moisture uptake of wood [50]. |
| 19, impregnated | Copper ethanolamine | Copper can cause membrane damage [46] and protein denaturation [47]. |

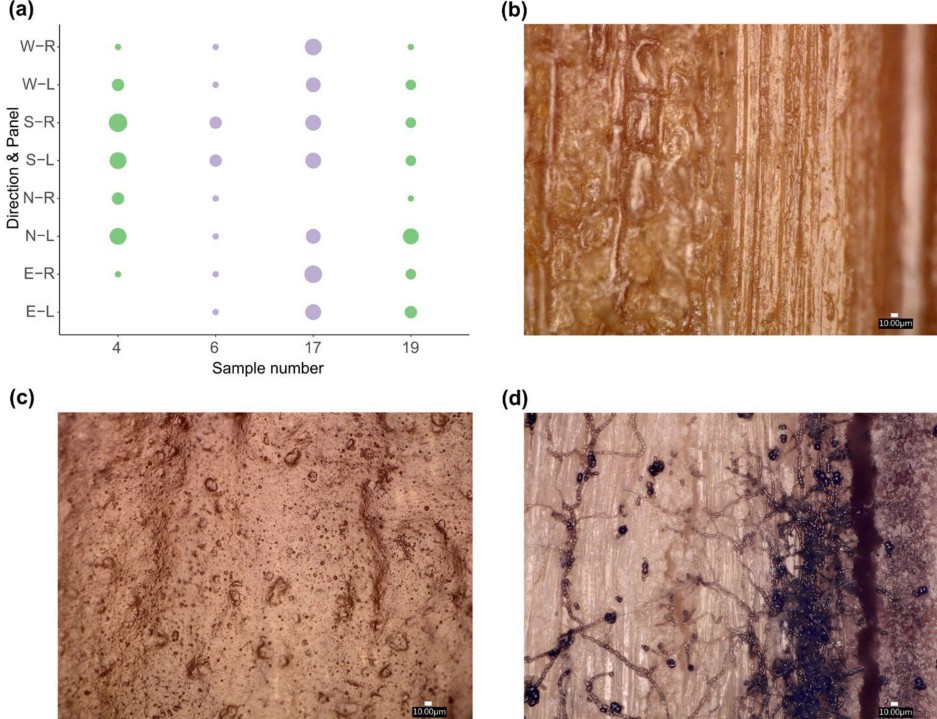

**Figure 3.** (**a**) Samples resistant to fungal growth. The abbreviations on Y-axis represent cardinal direction and panel (e.g., E-L = east left, N-R = north right). Colours represent coated (purple) and non-coated (green) samples. Symbol size corresponds to the number of CFUs (<10). (**b**–**d**) Microscope images of samples (**b**) 4, (**c**) 6, and (**d**) 13 at 500×. Scale bar represents 10 µm.

Sample 4 contains bamboo cladding and its resistance to fungi is surprising, given the breadth of the literature reporting its susceptibility to mould and decay [51]. However, the material used in this experiment (stick-glued bamboo façade profiles) provided by the industrial partner was additionally protected by oil, which might explain its high resistance. The second non-coated sample (19) was treated with copper ethanolamine, which is known to prevent fungal growth. For the two coated samples, 6 and 17, provided by industrial partners, the exact composition of the coating is unknown, and it is, therefore, difficult to interpret these results. It must be mentioned that only sample 19 from Table 1 demonstrated antifungal properties in practice, while we would expect similar performance from at least sample 10. However, the treatment of sample 10 was conducted under laboratory conditions

contrary to sample 19, which was prepared by an industrial partner. This might be the reason for the lower effectiveness of the treatment in this particular case.

### 3.4. Dominant Species

In long-term experiments, *Aureobasidium* sp., a polyextremotolerant fungus [52], has consistently been shown to be the prevalent species on wood-based materials [14,17,36]. Podgorski et al., who performed sampling every 3 months, reported that the diversity of fungal species decreases over time and that it is after 9 months that *Aureobasidium* sp. predominates [14]. To investigate the occurrence of dominant species in more detail and in a more quantitative manner, at the 1-month timepoint we identified samples where ≥ 90% of the CFUs appeared to have the same morphology. Pure cultures were obtained and selected DNA regions (internal transcribed spacers 1 and 2 including the 5.8S rDNA (ITS) and/or actin and/or β-tubulin) were sequenced to allow for identification. Dominant species were identified on at least one sample (out of 8) for 16 of the 33 tested materials (Figure 4a). Most of them were "heavily" colonised, having a CFU count of 100. On the vast majority of these materials, *Aureobasidium* sp. was the dominant genus, implying that *Aureobasidium* can establish dominance already during the early phases of colonisation. We found that *Aureobasidium melanogenum* (*A. melanogenum*) is the predominant species, with only a few occurrences of *A. pullulans*. However, it was only in 2014 that *A. melanogenum* was defined as a species [53]. Before this, it was a variety of *Aureobasidium pullulans* (*Aureobasidium pullulans* var. *melanogenum*), which explains why *A. pullulans* is the species commonly mentioned in publications. It has been shown that *A. melanogenum* is the predominant *Aureobasidium* species on oil-treated wood [54], but no other studies have, to our knowledge, compared the species composition on commercial cladding materials since *A. melanogenum* acquired the status of a species. *A. melanogenum* is considered an opportunistic pathogen as it can grow at 37 °C, which is a unique trait in the genus [53]. This might make it more competitive against *A. pullulans* when growing on surfaces exposed to warm weather. Another possible advantage of *A. melanogenum* could be its higher melanin content [55], which could provide efficient UV protection. Importantly, there might also be several other reasons contributing to the prevalence of *A. melanogenum*—such as its higher presence in the local environment, better attachment, faster growth, biofilm formation etc.

Only one material representing a wood–plastic composite (32) had a dominant species on all replicates. Different species of *Cladosporium*, as well as *Alternaria alternata* and *Penicillium* sp., were found on the surface of this material. The spores of all three species are commonly present in the outdoor air [56–59]. It must be mentioned that this wood–plastic composite had a specific topography (grooved surface), which might have influenced the retention of certain types of spores and/or led to non-homogenous growth over the material.

Representative macro- and micromorphologies of all species isolated from the materials are shown in Figure 4b,c, respectively. *A. melanogenum* is smooth, matt, and pinkish with areas of dark brown, indicating melanisation. On the microscopic picture, partly melanised vegetative hyphae and conidia can be observed. *A. pullulans* colonies are smooth, matt, and pinkish. Hyaline vegetative hyphae and conidia are visible. The colonies of *Cladosporium* sp. are powdery and olivaceous green, septate hyphae with conidiophores and conidia can be discerned from the microscopic image. *Alternaria alternata* appears woolly, olive-brown, and covered with short aerial hyphae. Dark, septate hyphae and ellipsoidal conidia, some in chains, are present. *Penicillium* sp. colonies appear powdery and greenish-blueish. Typical conidiophores with phialides forming basipetal conidia can be observed.

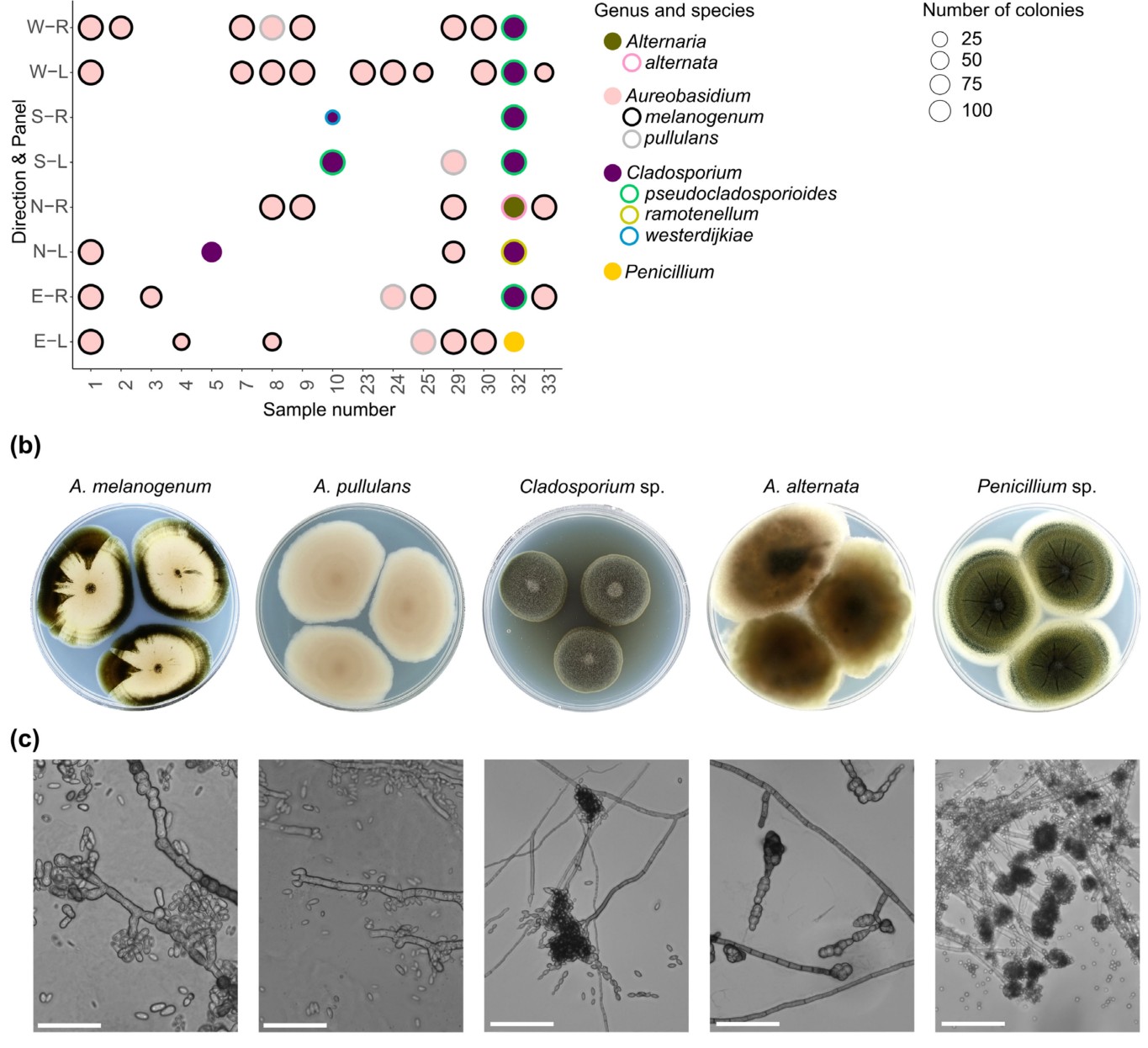

**Figure 4.** (**a**) Samples with dominant species. The abbreviations on Y-axis represent cardinal direction and panel. Colours represent fungal identity; inner symbol size corresponds to the number of CFU. (**b**) Images of pure cultures. (**c**) Microscopic images of pure cultures obtained with a 60× objective. Scale bar represents 50 μm.

### 3.5. Materials Characterisation

Roughness and wettability measurements for selected samples exposed to the north were conducted before and after 2.5 months of exposure. Samples were dismounted from the stand and stored in a controlled environment to reach equilibrium moisture content before measurements. The highest increase in surface roughness was evident for natural cladding materials (e.g., samples 29, 23, or 11). The arithmetical mean surface height (Sa) is the basic irregularity quantifier corresponding to the arithmetical mean height of the roughness profile (Ra), traditionally determined from the two-dimensional surface roughness profiles. ΔSa for the averaged measurements of samples 29, 23, and 11 was 8.9, 5.9, and 5.2 μm, respectively. An increase in surface roughness during weathering is associated with general erosion of the wood surface, removal of the single fibres, and the

leaching of photodegraded components [60]. Changes in the surface roughness during the weathering process are also linked to increased susceptibility to dirt accumulation, moisture, and pollutants as well as enhanced spore attachment and germination [61]. On the contrary, samples protected with a coating (e.g., 6) did not exhibit a change in the Sa parameter. Similar observations can be made while interpreting wettability results. Low contact angle values indicate that the liquid easily spreads over the assessed surface. Conversely, a high contact angle implies poor spreading and physical affinity. Unprotected samples already exhibited lower contact angle measurements after 2.5 months of the natural weathering process. The surfaces were easily wetted, and the sessile drops disappeared after a few seconds of the test. Contact angle parameters remained stable for coated samples (e.g., 6) where the difference was negligible. The contact angle measurement implies wettability by water (or other liquids) of the exposed material surface. The extent to which the contact angle changes is an important indicator of the progression of weathering [62]. Evaluation of investigated biobased cladding materials confirms that the most common material characteristics for improving bioreceptivity are high surface roughness, high open porosity, high capillary water content, and high wettability [63].

## 4. Conclusions

The goal of this work was to examine the early fungal colonisers on a diverse set of façade materials including commercial, coated, and non-coated materials. A set of 33 wood-based cladding materials were exposed to four cardinal directions and monitored in outdoor conditions. The response of these cladding materials to weathering and fungal infestation was assessed to verify if fungal presence is necessarily connected with material deterioration. The surfaces were sampled with a wet swab and plated on DG-18 agar. Pure cultures were then isolated and identified through PCR amplification and Sanger sequencing of specific DNA regions/genes.

- On most of the investigated materials where a dominant morphology was present, *Aureobasidium* sp. was the dominant genus, implying that *Aureobasidium* can establish dominance already during the early phases of colonisation.
- Both the material type and the climate condition at the exposure site influenced fungal colonisation. Samples exposed in another climate zone or at a different time of the year might exhibit different infestation patterns. Therefore, the current findings can only be interpreted within the experimental setting used.
- Coated materials were generally less susceptible to fungal infestation. The distributions of the CFU counts on samples exposed in four directions differed significantly. The least growth was observed in the south, while the highest fungal burden was observed in the east and west.
- Based on described results, *Aureobasidium* sp. is considered a candidate for a living component of a new nature-inspired coating system designed to effectively protect architectonic surfaces. We assume that due to its polyextremotolerance, *Aureobasidium* sp. would be a good candidate for living coatings to be used in different climate zones. Nonetheless, further research in diverse climatic regions might identify alternative species and/or even polymicrobial communities as candidates for living coatings.

**Author Contributions:** Conceptualisation, K.B.O., A.G., F.P. and A.S.; methodology, K.B.O., A.G., F.P. and A.S.; software, K.B.O. and A.G.; validation, K.B.O., A.G., F.P. and A.S.; formal analysis, K.B.O., A.G., F.P. and A.S.; investigation, K.B.O., A.G., F.P. and A.S.; resources, A.S.; data curation, K.B.O.; writing—original draft preparation, K.B.O.; writing—review and editing, K.B.O., A.G., F.P. and A.S.; visualisation, K.B.O.; supervision, A.S.; project administration, A.S.; funding acquisition, A.S. All authors have read and agreed to the published version of the manuscript.

**Funding:** Part of this work was conducted during the project WoodLCC (#773324), which is supported under the umbrella of ERA-NET Cofund ForestValue by the Ministry of Education, Science and Sport (MIZS)—Slovenia. The authors gratefully acknowledge the European Commission for funding the InnoRenew project (Grant Agreement #739574) under the Horizon2020 Widespread-Teaming

program, and the Republic of Slovenia (investment funding from the Republic of Slovenia and the European Union's European Regional Development Fund). This research was co-funded by the European Union (ERC, ARCHI-SKIN, #101044468). Views and opinions expressed are, however, those of the author(s) only and do not necessarily reflect those of the European Union or the European Research Council. Neither the European Union nor the granting authority can be held responsible for them.

**Institutional Review Board Statement:** Not applicable.

**Informed Consent Statement:** Not applicable.

**Data Availability Statement:** The dataset used for analysis in this study is available at the Zenodo.org Open Access Repository [64]. Created with BioRender.com, accessed on 4 August 2023.

**Acknowledgments:** The authors would like to acknowledge Nina Gunde-Cimerman for enabling access to the equipment and providing materials necessary for isolating DNA and performing PCR reactions, Mojca Matul for her help with depositing and retrieving the strains from the Microbial Culture Collection Ex, and Polona Zalar for her advice.

**Conflicts of Interest:** The authors declare no conflict of interest. The funders had no role in the design of the study; in the collection, analyses, or interpretation of data; in the writing of the manuscript; or in the decision to publish the results.

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
