# Peer review of "Assessing the Bioreceptivity of Biobased Cladding Materials"

_coatings, doi:10.3390/coatings13081413_

Round 1

Reviewer 1 Report

1. The author is very specific about the experiments and the analytical methods; however, the downside is that there are no pictures of the overall experimental process, and it is recommended that beautiful pictures be added.

2. What does the 217 before sample #23 in Table 1 represent?

3. The labels on the pictures are suggested to be consistent with the journal format.

4. Concerning the conclusion, I hope it can be elaborated in points.

Reviewer 2 Report

The manuscript entitled ‘’Assessing the bioreceptivity of biobased cladding materials’’ considers full-scale tests of coating samples in the autumn period. There is a definite novelty. A large biological experiment was carried out by the authors. On the one hand, a outdoor experiment is an advantage because it concerns real façade materials, and on the other hand, such an experiment is not universal. Nevertheless, the study of the authors is of some interest and worthy of publication.

It should be noted a large scatter in the contact angle (Figure 1) for samples 7, 10, 12, 23 and 24. It is unlikely that they correspond to p ˂0.05.

In the Conclusion section, it should be mentioned that the results are obtained for a certain climatic region. Perhaps in other regions, different microorganisms will be better candidates for new nature-inspired coating systems.

Reviewer 3 Report

The publication provided for review concerns the determination of the bioresistance of selected coatings. I found the topic interesting and still relevant. The authors examined 33 wood-based materials exposed to various factors. Biological material was isolated from the surface of these materials. At the same time, the resistance of the tested materials to the analysed agents was determined.

The publication is written in a standard structure. The introduction reflects the research topic, the methodology describes the research performed, and the description of the research results is consistent with the results presented. The conclusions are related to the paper's subject matter, although the authors indicate that this is an introduction to a more extended study.

Specific comments:

- The aim of the paper should present only the assumptions of the research and the research thesis. It should not describe the conduct of the study.

- Table 1 should be in the methodology section.

- When the support material is MDF, it is better to give fibreboard or MDF rather than composite.

- The publication is quite long, but extending it with a more detailed description of the prepared samples would be worthwhile.

- Was the wetting angle also measured immediately after the samples were removed from the roof, or only after a seasoning period?

- How did the moisture content of the samples develop if the air humidity ranged from around 50% to over 90% during the test period? Did this not affect the roughness of the material after the tests?

- The 'Conclusion' chapter should be rewritten only to answer the objectives and questions asked in the paper.

Overall, the paper is written in a readable style. The figures and photos do not need any corrections. The discussion of results refers to other studies. Some of the comments above can be taken as discussion.
